# Hyperuricemia and Endothelial Function: Is It a Simple Association or Do Gender Differences Play a Role in This Binomial?

**DOI:** 10.3390/biomedicines10123067

**Published:** 2022-11-29

**Authors:** Tiziana Ciarambino, Pietro Crispino, Mauro Giordano

**Affiliations:** 1Internal Medicine Department, Hospital of Marcianise, ASL Caserta, 81035 Caserta, Italy; 2Internal Medicine Department, Hospital of Latina, ASL Latina, 04100 Latina, Italy; 3Department of Advances Medical and Surgical Sciences, The University of Campania Luigi Vanvitelli, 80138 Naples, Italy

**Keywords:** vascular endothelium, cardiovascular events, uric acid levels, sex-related differences

## Abstract

The endothelium plays a fundamental role in the biological processes that ensure physiological vessel integrity, synthesizing numerous substances that are capable of modulating the tone of vessels, inflammation and the immune system, and platelet function. Endothelial dysfunction refers to an anomaly that develops at the level of the tunica that lines the internal surface of arterial and venous vessels, or, more precisely, an alteration to normal endothelial function, which involves the loss of some structural and/or functional characteristics. Studies on sex differences in endothelial function are conflicting, with some showing an earlier decline in endothelial function in men compared to women, while others show a similar age of onset between the sexes. Since increased cardiovascular risk coincides with menopause, female hormones, particularly estrogen, are generally believed to be cardioprotective. Furthermore, it is often proposed that androgens are harmful. In truth, these relationships are more complex than one might think and are not just dependent on fluctuations in circulating hormones. An increase in serum uric acid is widely regarded as a possible risk factor for cardiovascular disease; however, its role in the occurrence of endothelial dysfunction has not yet been elucidated. Several studies in the literature have evaluated sex-related differences in the association between elevated uric acid levels and cardiovascular events, with conflicting results. The association between uric acid and cardiovascular disease is still controversial, and it is not yet clear how gender differences affect the serum concentration of these substances. This review was primarily aimed at clarifying the effects of uric acid at the level of the vascular endothelium and describing how it could theoretically cause damage to endothelial integrity. The second aim was to determine if there are gender differences in uric acid metabolism and how these differences interact with the vascular endothelium.

## 1. Background

A physiologically healthy endothelium is characterized by a balance between factors with a vasoconstrictive function and factors with a vasodilatory function. This balance is responsible for the integrity of the vascular walls. On the contrary, the breaking of this equilibrium involves a decline in endothelial function and precedes the development of overt cardiovascular diseases. The endothelium plays this decisive role due to its ability to properly synthesize nitric oxide (NO), which, according to the needs of our organs, is capable of inducing the release of vascular endothelial cells in response to a vasodilatory stimulus [1].

Uric acid is the final product of purine metabolism, which is initiated by the catalytic activity of xanthine oxidase, leading as an intermediate metabolic product to the production of reactive oxygen species [2]. Acute and chronic kidney injury and a hyperproteic diet influence purine concentration while a low protein diet ameliorates purine metabolism according to a mechanism by gender [3,4,5,6,7]. Among these catabolic products, superoxide can cause the weakening and deterioration of endothelial function through its binding with nitric oxide (NO), limiting the bioavailability of this important vasodilator. This process would seem to attribute to uric acid a role as a cardiovascular risk factor [8,9].

However, it has not yet been determined whether high concentrations of uric acid are dangerous for endothelial integrity and whether concomitant factors are needed to identify a risk to the cardiovascular system. Experimental studies have indicated that uric acid causes endothelial dysfunction through an increase in oxidative stress and, therefore, the presence of a persistent inflammatory state [10]. Such a mechanism, however, has not yet been confirmed to take place in vivo, as it is difficult to study the effects of uric acid on the vascular endothelium in isolation; furthermore, it has not yet been clarified whether this mechanism works in synergy with other cardiovascular risk factors to contribute to vascular damage. Recent epidemiological studies have also shown that an increase in uric acid concentration is significantly associated with the onset of arterial hypertension, dyslipidemia, diabetes mellitus, chronic renal failure, and atrial fibrillation [11,12,13,14], as well as an increased possibility of cardiovascular events [15].

It has long been known that serum uric acid levels are different in men and women and that this diversity is linked to the differing hormonal characteristics of the two sexes. An association between serum uric acid levels and the risk of vascular disease has previously been reported [16,17]. In particular, two studies [9,17] showed that an increase in serum uric acid constituted a risk factor for stroke in female but not in male subjects. However, the role of sex hormones in influencing the health status of the cardiovascular system is significantly more complex and is affected by age as well as sex differences in exposed tissues. Although so far in the literature much importance has been placed on the role of gender in endothelial function and, separately, uric acid metabolism, no attention has been paid to how gender affects the activity of the metabolic products of uric acid in relation to endothelial function.

## 2. Methods

Clinical trials published on or before 30 September 2022 were identified by Pubmed. The search keywords were gender/sex differences, endothelial dysfunction, cardiovascular risk, hyperuricemia, nitroxide, xanthine oxidase, NO synthase (eNOS), estrogen, androgens, progesterone, and testosterone. The references of the selected studies were reviewed for potential inclusion. Studies written in languages other than English were excluded. Two authors (P.C. and T.C.) reviewed all study abstracts. All selected studies were qualitatively analyzed.

## 3. Endothelial Dysfunction and Hyperuricemia

The endothelium is a metabolically active barrier system made up of cells that produce various molecules involved in determining changes in vascular tone. The most important of these molecules, as highlighted above, is nitric oxide (NO), but other products of endothelial metabolic function include prostaglandin I2; the endothelium-derived hyperpolarizing factor; and vasoconstricting agents such as endothelin-1, thromboxane A2, and angiotensin II [18,19]. The balance between these factors guarantees normal vascular tone. However, the endothelium also plays an important role in preventing events that could have a pro-thrombotic, pro-inflammatory, and pro-oxidative effect. The main cardiovascular events, however, largely depend on the atherosclerotic effects linked to a decrease in NO production, the loss of the ability to replace vascular smooth muscle cells, a lack of leukocyte adhesion inhibition, and platelet aggregation and adhesion [20]. Endothelial function is impaired in patients with hyperuricemia, but it has not been fully determined whether hyperuricemia itself is a causal risk factor for endothelial dysfunction.

The main mechanisms by which uric acid causes endothelial dysfunction are related to the decoupling of xanthine oxidase and endothelial NO synthase (eNOS) and to the altered functioning of uric acid transporters.

Firstly, the activity of xanthine oxidase is greater when it is necessary to metabolize excess purine residues in the plasma. Such enzymatic activity generates reactive oxygen species (ROS), which could have a deleterious effect on endothelial function. The ROS react directly with NO, creating high-affinity bonds that reduce the bioavailability of NO through its degradation and inactivation and increase the formation of peroxynitrite, which is capable of causing DNA damage, cell death, and lipid peroxidation. This process is called eNOS decoupling [21]. Xanthine oxidase is a protein with an enzymatic function that can be converted into the other form of xanthine oxidoreductase, namely xanthine dehydrogenase (XD). XD preferentially uses nicotinamide adenosine diphosphate (NAD+) as an electron acceptor to convert hypoxanthine to xanthine and xanthine to urate, generating nicotinamide adenosine dehydrogenase (NADH), while XO preferentially uses molecular oxygen as an electron acceptor to oxidize purines, leading to the generation of superoxide anions (O_2_^−^) and hydrogen peroxide (H_2_O_2_). Therefore, under conditions in which the activity of XO is increased to metabolize an excessive plasma load of purines, uric acid and other reactive oxygen species (ROS) are generated, which have a deleterious effect on endothelial function. The excessive production of oxygen free radicals in conjunction with the increased production of uric acid leads to the free radicals reacting directly with NO with a high affinity, resulting in the reduced bioavailability of NO through its degradation and inactivation and the increased formation of peroxynitrite (ONOO−). XO exists not only within the cytoplasm of endothelial cells but also on the outer surface of the endothelial cell membrane. Furthermore, some studies have shown that high levels of circulating XO are released by organs and tissues where such enzymes are abundant in pathological conditions. These results suggest that XO is not only active in endothelial cells, but also present in its free form, representing a source of ROS that contribute to endothelial dysfunction [1]. In addition to the decoupling effect on nitric oxide, high xanthine oxidase activity has been observed in concomitance with the transformation of macrophages into foam cells [22]. This event represents the *primum movens* of all the processes that occur locally in the endothelium and cause the development of plaque and its complications [23,24]. At this point, the question arises as to how xanthine oxidase participates in the transformation of macrophages into foam cells and why it plays an important role in the processes of atherosclerotic plaque formation. A previous histological study found that xanthine reductase is also present in macrophages, above all in those clustered near the aortic root wall [25]. Macrophages play a key role in the development of atherosclerosis by moving into pathological areas (such as those common in the dysfunctional endothelium) and, in the presence of lipid deposits, transforming into foam cells. The conversion of macrophages into foam cells increases the activity of the xanthine oxidase present inside these cells. Furthermore, studies have described xanthine oxidase as an endogenous regulator of cyclooxygenases, and it would therefore play an important role in the regulation of inflammatory response and the functioning of the innate immune response [26].

As regards the cellular transport system of uric acid, it is known that in humans, almost all uric acid is filtered by the glomeruli and reabsorbed from the renal tubules through transporter subunits present on the surface of the glomerular lumen (URAT 1), before reaching the basolateral membrane where the URATv1 transporter is located, which plays an important role in uric acid absorption in endothelial cells [27,28]. Experimental studies have shown that uric acid transporters are also expressed in vascular endothelial cells, contributing to inflammation; oxidative stress; and, hence, endothelial dysfunction through a reduction in endothelial NO [29,30]. Recent experimental studies suggest that uric acid absorbed into endothelial cells via URATv1 causes endothelial dysfunction by reducing the bioavailability of NO through increased inflammation and oxidative stress and the decreased phosphorylation of eNOS due to the activation of NADPH oxidase [31,32]. In addition to increased oxidative stress, previous studies have indicated that UA is involved in the regulation of endothelial function through other mechanisms. The role of uric acid in increasing the activity of arginase and, therefore, reducing the availability of the L-arginine that is required by the endothelium for the generation of NO is important. In addition, AU also promotes the reactivity or instability of the C-protein due to the increase in endothelial free radicals with a further decrease in NO production [33]. 

## 4. Effect of Sex Hormone Differences on Endothelial Function

Male individuals develop cardiovascular diseases attributable to endothelial dysfunction earlier and more frequently than women. This difference between the two sexes persists until menopause, when endothelial dysfunction has a slightly higher incidence in women than in men [34,35]. The decline in female sex hormones, especially estrogen, and the negative effect of androgens are the factors that underlie the loss of the protection enjoyed by women of childbearing age.

In women, estrogen receptors (ERs) promote NO release via the endothelial activation of NO synthase (eNOS). Conversely, androgen receptor (AR) binding and activation can result in impaired endothelial NO release [36,37]. Indeed, in men, androgenic stimulation can induce increases in blood pressure through a NO-lowering mechanism. Apart from the effects on blood pressure in men, androgens also play a protective role on the cardiovascular system, both directly and indirectly through conversion into estrogen [38,39].

In men, bioavailable testosterone can exert its effects directly on androgen receptors (ARs). Alternatively, it may be metabolized to other steroid hormones, such as dihydrotestosterone (DHT) or 17β-estradiol (E2), or by 5α-reductase and aromatase, respectively [40]. In women, progesterone exerts its most important effects on the cardiovascular system, inducing the release of mediators such as NO and cyclooxygenase [10,41,42,43,44]. Progesterone also affects the smooth muscle cell tone of vessels by inducing a rapid increase in calcium ions. Therefore, considering the effects on smooth muscle cells, it is possible to observe that progesterone performs vasoconstrictive activity in some areas, while in others it exerts vasodilatory activity.

Estrogens significantly contribute to the regulation of endothelial function through mechanisms that are not yet well understood. It is known that some subtypes of estrogen re-receptors are expressed at the endothelial level and on smooth muscle cells, but the role of each of these subtypes is not yet known [45,46]. The number and stimulation of these receptors in various organs of the human body depend on the type of tissue and are sensitive to circulating estrogen levels [47,48,49,50]. The E2 receptors, if stimulated, can also induce a rapid release of eNOS, which encourages a consideration of the concomitant presence of non-genomic mechanisms in the functioning of these receptors [10,47,51]. Estradiol can also increase sensitivity to vasodilatory factors, such as acetylcholine or prostaglandins, thus reducing the concentrations required to evoke similar vasodilatory responses [52,53]. In women of childbearing age, estradiol is associated with an increase in vascular relaxation, mainly mediated by NO [54,55]. Ethinylestradiol also attenuates vasoconstrictive responses to the infusion of noradrenaline in perimenopausal women [10]. However, after menopause, the in-adrenergic receptor contributes to a reduction in vascular tone, which increases the risk of hypertension in postmenopausal women [56]. Thus, the mechanisms contributing to greater E2 receptor-mediated vascular relaxation in women include increased NO bioavailability, which is associated with increased beta-adrenergic and decreased alpha-adrenergic sensitivity. Men express ER receptors at the endothelial level that appear to exert cardioprotective functions similar to those of females [57]. In men, E2-mediated vasodilation causes an increase in the concentration of prostacyclin and NO within minutes [58]. The main source of E2 in men is the conversion of testosterone by the aromatase present in smooth muscle cells. Hence, the vasodilatory effect in males depends on the circulating E2 and, indirectly, on the concentrations of both testosterone and aromatase [59]. Estradiol can also have a cardioprotective effect in men by limiting the proliferation and migration of vascular smooth muscle cells through mechanisms mediated by ER-alpha and ER-beta receptors [60,61]. Conversely, increased exposure to ER-beta and E2 causes early atherosclerosis in the coronary arteries in humans, suggesting a dual role, especially for E2, in endothelial dysfunction [62].

Gender differences in the functioning of the estrogen receptor ET-1 are known to be common. ET-1 is an enzyme present in two forms and acts mainly through an autocrine or paracrine mechanism signaling at the level of the tissue in which it is synthesized. The vasoconstrictive effects of ET-1 are mediated by the binding of ET-1 to ET type A, whereas binding to ET type B induces the release of endothelium-derived vasodilators such as NO. Meanwhile, ET-1 mediates vascular tone and the equilibrium mediated by the two isoforms ETAR and ETBR. The ET-1 receptor also induces the production of growth factors, ROS, and inflammatory mediators [63,64,65]. Available studies indicate that ET-1 levels fluctuate more in women based on the concentration of circulating female hormones compared to men of the same age [66,67]. This fluctuation depends mainly on the effect of the luteal phase of the menstrual cycle in women of a fertile age, when hormones are elevated. The data from these studied suggest that ET-1 production in the vascular endothelium is attenuated in the presence of estrogens and progesterone. Complementary to these sex-hormone-mediated attenuations in ET-1 synthesis, increases in endothelial ETBR protein expression with estrogen and progesterone may also increase the clearance of circulating ET-1 [68,69]. In general, women have a greater ETBR protein expression compared to men, and these differences likely also contribute to sex differences in circulating ET-1. These sex differences in vascular responsiveness to ET-1 are mediated, at least in part, by differences in the expression and location of ET-1 receptor subtypes. 

## 5. Hyperuricemia and Endothelial Dysfunction and Gender

The clinical and epidemiological data described so far show that NO production is the basis of the relationship between endothelial damage, UA metabolism, and gender. We have so far pointed out how impaired endothelial function, characterized by an imbalance in the endothelial activity of nitric oxide synthase (eNOS), precedes and accelerates the development of acute cardiovascular events. At this point, however, one must question the specific role of NO: Is it only an agent that induces vasodilation in response to a stimulus, or is it a determining element that performs even more important functions? If these other functions are present, what role do gender differences play in AU metabolism? To answer these questions, we must consider hereditary gender differences and acquired gender differences.

### 5.1. Hereditary Sex-Specific eNOS Activity and Function in Human Endothelial Cells

Studies on the human genome have shown that many thousands of genes coding for functional proteins in our bodies and in particular enzymes are differentially expressed in men and women [70]. Additionally, with regard to the physiology of endothelial cells, some authors have highlighted traits that testify to gender differences [71,72]. It would appear that male endothelial cells show greater reactivity to vascular stress both in mechanical terms and with regard to immune response than their female counterparts [72]. Endothelial dysfunction is triggered by reduced endothelial nitric oxide synthase (eNOS) activity and the consequent decrease in the availability of nitric oxide (NO). The enzyme eNOS represents the main source of NO production in endothelial cells, where it plays an important function not only in controlling the integrity of the endothelial barrier but also in modulating leukocyte adhesion, platelet aggregation, and angiogenesis [21,73]. Since females are more protected than males against major cardiovascular events, it has long been assumed that estrogen is able to genetically mediate and regulate the expression of eNOS in endothelial cells [47,74] Furthermore, the activation of the enzyme and the formation of NO can also occur through rapid signaling pathways mediated by other factors present in endothelial cells [75]. This collateral activity is also greater in premenopausal women than in men and is maintained even after menopause with estrogen-based replacement therapy [76,77]. More specifically, a study showed that female endothelial cells have a higher eNOS expression than their male counterparts [78]. Taking this into account, the authors hypothesized that differences in fetal and/or maternal levels could be responsible for the increased expression of eNOS in female endothelial cells, although this was ruled out by another study that demonstrated that the levels of 17β-estradiol are comparable between male and female fetuses [79]. Therefore, the increased congenital expression of female eNOS appears to be independent of exposure to and concentrations of estrogens, although the possibility that the high values of eNOS present in female endothelial cells indicate downstream epigenetic regulation has not been excluded [80]. It is interesting to note that the differential expression of genes can be correlated to different levels of reactivity to damage, resulting in the varied expression of immune genes and the greater ability to induce the differentiation of T cells thanks to estrogens from the maternal environment, and this may contribute to the protection against cardiovascular disease of the premenopausal female population [78,79]. Endothelial cell migration is essential for angiogenesis. This mobile process is directionally regulated by chemotactic and mechanotactic stimuli and involves the degradation of the extracellular matrix to allow the progression of migratory cells. It requires the activation of several signaling pathways that converge on the remodeling of the cytoskeleton [80]. eNOS is involved in migration and angiogenesis by modulating vascular endothelial growth factor (VEGF) function or angiogenesis damage-induced ischemia [48,49]. Additionally, with regard to this function, eNOS is more active in promoting the migration of endothelial cells in females than males [81]. Overall, it appears that eNOS is the main enzyme pathway used by endothelial cells to promote new angiogenesis, while in men it seems that this activity occurs independently of the expression and activity of eNOS [78]. More recently, it has been shown that alterations in the lipid component of the basement membranes of endothelial cells affect the function of eNOS, although to date no gender differences have emerged regarding this phenomenon [82].

### 5.2. Acquired Sex-Specific eNOS Activity and Function in Human Endothelial Cells

Hyperuricemia is caused mainly by abnormal liver metabolism, urate excretion, or rapid cell turnover. Renal excretion accounts for approximately two thirds of total urate excretion, and the remaining third is excreted mainly through the intestinal tract. Therefore, the influence of sex hormones, visceral adiposity, and muscle mass undergo specific changes over time that differ between the two sexes and contribute to both uric acid metabolism and endothelial damage. Changes in the hormonal balance and the increased metabolism of uric acid damage the vascular endothelium, largely due to the reduced availability of nitric oxide, increased oxidative stress, and the promotion of a state of chronic inflammation [83,84,85,86]. Vascular smooth cell proliferation, vascular inflammation, or atherosclerosis can be induced or accelerated by lipid peroxidation and platelet adhesion related to uric acid levels [87,88], but they may also result from the effect of sex hormones on endothelial function [10]. Condensing the contents of numerous observational studies has shown that increased UA levels induce endothelial damage in conjunction with hypertension, diabetes mellitus, and metabolic syndrome. Therefore, for the prevention of endothelial damage induced by UA, it is also necessary to adequately control vascular diseases such as arterial hypertension and other metabolic diseases such as diabetes mellitus, conditions that can potentially enter into synergy with each other [68]. Hormone levels undergo contraction changes largely related to age, which can have a huge impact on the development of metabolic diseases in men and women [89]. The concentration of UA in the population groups most affected by cardiovascular diseases related to endothelial dysfunction varies between elderly men and postmenopausal women [89], which indicates that gonadal hormones can play a decisive role. As for men, a study reported that elevated UA levels are associated with low levels of testosterone and sex hormone binding globulin (SHBG) [87], a protein produced in the liver that is capable of binding testosterone, dihydrotestosterone, and estradiol (estrogen) to transport them in an inactive form into the bloodstream. This general finding was clarified by another study, which demonstrated that a lack of male sex hormones linked to hypogonadism correlates with higher values of UA in diabetic subjects compared to those with normal glucose metabolism [90]. Furthermore, dehydroepiandrosterone (DHEA) has been recognized as an effective element due to its protective effects on metabolism against obesity, diabetes, and atherosclerosis [91]. Among the vascular-protective effects of DHEA, an increase in the concentrations of nitric oxide released by endothelial cells has been observed [92]. Furthermore, other studies have indicated that lower-than-normal DHEA values are linked to a higher incidence of conditions that represent cardiovascular risk factors [93]. On the other hand, in postmenopausal women, testosterone administration has been reported to cause a marked increase in plasma UA levels [87]. In the postmenopausal state, a significant increase in serum UA concentrations was observed, which decreased significantly with hormone replacement therapy [85,94]. Among the hormones used, it was found that only progestogens are able to sensitively reduce the levels of UA, while the intake of E2 is not associated with a decrease in UA in women [95,96]. The neutrality of E2 at the circulating levels of UA was also observed in male subjects [94]. However, it is possible that estradiol could reduce urate levels in both males and females by acting through other mechanisms. It is known that UA concentrations not only depend on its products but also on its elimination. Urate secretion and reabsorption in both renal tubules and the intestinal tract are regulated by proteins called urate transporters, among which ABCG2 is currently recognized as the most important transporter for extrarenal urate excretion [96]. A study administering estradiol benzoate (EB) to hyperuricemic male mice and female mice after the removal of the ovaries confirmed the urate-lowering effect of estradiol through the upregulation of intestinal ABCG2 expression. It was hypothesized that estradiol might also upregulate intestinal ABCG2 to promote intestinal urate excretion. This led to the hypothesis that in both men and women estradiol can promote the intestinal excretion of urate by upregulating the intestinal ABCG2 transporter, leading to a decrease in blood circulation [96].

## 6. Conclusions

The main conclusions of this study are summarized in Table 1. Experimental and clinical studies have shown that endothelial dysfunction is caused by increased ROS generation in the process of purine metabolism catalyzed by xanthine oxidase or is dependent on the direct action of sex hormones on the receptors expressed by endothelial cells. However, it has not been fully determined whether uric acid and sex hormones interact to cause endothelial dysfunction in humans. Although experimental studies have shown that uric acid absorbed into endothelial cells via the activation of uric acid transporters causes the major accumulation of nitroxide, no clinical study has shown that sex hormones interact with this transporter system and influence endothelial function. Gender differences in the evolution of endothelial damage have been reported in the past. Hormone levels have a particular impact in this regard, influencing the production and activity of NO and providing women with greater protection than men against acute cardiovascular events. Estrogen also plays a major role in regulating the serum levels of uric acid, reducing its accumulation in the walls of blood vessels and therefore limiting endothelial damage in women compared to men. In men, androgen stimulation can induce an increase in blood pressure through a nitroxide-reducing mechanism, although androgens also play a protective role in the cardiovascular system through both direct and indirect (via conversion to estrogen) action on the vascular endothelium. In humans, bioavailable testosterone can exert its effects directly on androgen receptors (ARs). Alternatively, it can be metabolized to other steroid hormones, such as dihydrotestosterone (DHT) or 17β-estradiol (E2), or by 5α-reductase and aromatase, respectively. It is well known that testosterone levels decrease and cardiovascular mortality increases with age, but the association between testosterone and cardiovascular disease remains unclear. On the one hand, estrogen promotes the release of NO; on the other hand, it prevents the production of oxygen free radicals and the chronic inflammatory state deriving from the metabolism of uric acid.

## Figures and Tables

**Table 1 biomedicines-10-03067-t001:** Take-home messages.

Endothelial dysfunction. Is caused by increased ROS produced by xanthine-oxidase activity or is dependent on the direct action of sex hormones on the receptors exhibited by endothelial cells.
Estrogen promotes the release of NO and on the other hand, prevents the production of oxygen free radicals and the chronic inflammatory state deriving from the metabolism of the uric acid.
Estrogens play a major role in regulating serum levels of uric acid, resulting in less accumulation in the walls of blood vessels and therefore limiting endothelial damage compared to men.
In men, androgen stimulation plays a protective role in the cardiovascular system with both a direct and secondary action on the vascular endothelium by converting to estrogen.
Menopause and andropause increase uric acid levels and endothelial damage, but the association of this phenomenon with cardiovascular mortality remains unclear.

## Data Availability

Not applicable.

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
