# Peer review of "Hyperuricemia and Endothelial Function: Is It a Simple Association or Do Gender Differences Play a Role in This Binomial?"

_biomedicines, 2022, doi:10.3390/biomedicines10123067_

Round 1
Reviewer 1 Report
In the present manuscript, the authors provide a review of hyperuricemia and its relationship with endothelial dysfunction. In addition, they have attempted to describe a possible relationship of gender with this product, to date poorly studied this relationship. Although this review is quite complete, I would like to add several comments that might help the understanding and improvement of this manuscript.
1. The authors should make a drawing showing the endothelial cells displaying its functionality and the molecules involved (NOS and XO).
2. The authors should elaborate on the underlying mechanisms of endothelial dysfunction due to hypuricemia.
3. Is there any relationship of hyperuricemia and endothelial dysfunction associated with chronological age? Have the authors found any scientific article where this relationship is reported?
Author Response
4 November 2022
Dear Prof. Doctor
Please, find enclosed the revised version of the manuscript entitled: “HYPERURICEMIA AND ENDOTHELIAL FUNCTION: IS IT A SIMPLE ASSOCIATION OR IS THERE A ROLE FOR GENDER DIFFERENCES IN THIS BINOMIAL?” We thank the Editor and the Reviewers for their comments and we hope that the following changes will now make the manuscript suitable for publication on the Biomedicines. Please see the following list of the changes made in manuscript.
Reviewer 1
- In accordance with the reviewer's suggestion, a drawing showing the endothelial cells showing their functionality and the molecules involved (NOS and XO) was included in the work and cited in the text.
- According to with the reviewer's suggestion the part on the mechanisms underlying endothelial dysfunction due to hyperuricemia has been detailed and better dealt with in the text.
- Accepting the auditor's solicitation, indeed, in the literature we found works that describe either the variations by age group in serum uric acid levels or underline the fact that persistently elevated plasma uric acid values ​​worsen endothelial damage in the elderly and associated with other cardiovascular risk factors such as diabetes, hypertension arterial and dyslipidemia in contributing to atherosclerosis. We retraced the study of the literature, inserting various items, but at the end of this path, we did not find a work that directly activates the variations in uric acid levels as a function of endothelial damage.
Best regards
Tiziana Ciarambino
MD, PhD

Reviewer 2 Report
The present review describes in detail the mechanisms of the pathological effect of uric acid on the functional state of the vascular endothelium and discusses possible causes of gender differences. There are few comments:
1) There is no discussion of the work Albrecht E. et all. «Metabolite profiling reveals new insights into the regulation of serum urate in humans» (doi: 10.1007/s11306-013-0565-2), in which it is shown that of the 38 main metabolites of uric acid, 25 have significant sex differences.
2) Parts 4.1 (Testosterone), 4.2 (Progesterone), 4.3 (Estrogens), 4.4 (Gender differences in ER-receptor) seem to be excessive. These sections provide well-known data, but do not show the relationship between hormones and hyperuricemia. These sections should be significantly reduced in the volume of the chapter 4 «SEX HORMONES DIFFERENCES IN ENDOTHELIALFUNCTION»
3) It is necessary to conduct a thorough revision of the reference list. For example, work Wan H. et all. «The Associations Between Gonadal Hormones and Serum Uric Acid Levels in Men and Postmenopausal Women With Diabetes» is mentioned three times - â„–80 (line 550), â„–84 (line 559) и â„–89 (line 572).
Author Response
4 November 2022
Dear Prof. Doctor
Please, find enclosed the revised version of the manuscript entitled: “HYPERURICEMIA AND ENDOTHELIAL FUNCTION: IS IT A SIMPLE ASSOCIATION OR IS THERE A ROLE FOR GENDER DIFFERENCES IN THIS BINOMIAL?” We thank the Editor and the Reviewers for their comments and we hope that the following changes will now make the manuscript suitable for publication on the Biomedicines. Please see the following list of the changes made in manuscript.
Reviewer 2
1) Accepting the reviewer's suggestion, the work of Albrecht E. et al. "The metabolic profile reveals new knowledge on the regulation of serum urate in humans" (doi: 10.1007 / s11306-013-0565-2), was mentioned and included in the references.
2) Accepting the reviewer's suggestions, parts 4.1 (Testosterone), 4.2 (Progesterone), 4.3 (Estrogen), 4.4 (Gender differences in the ER receptor) have been reduced and associated with chapter 4 "SEXUAL HORMONE DIFFERENCES IN ENDOTHELIAL FUNCTION"
3) Accepting the auditor's suggestion, a thorough review of the reference list was carried out. In particular, the work of Wan H. et al. "The associations between gonadal hormones and serum uric acid levels in men and postmenopausal women with diabetes" is mentioned as reference â„–80, eliminating subsequent repetitions.
Best regards
Tiziana Ciarambino
MD, PhD
